# Transfer-Learning-Based Intrusion Detection Framework in IoT Networks

**DOI:** 10.3390/s22155621

**Published:** 2022-07-27

**Authors:** Eva Rodríguez, Pol Valls, Beatriz Otero, Juan José Costa, Javier Verdú, Manuel Alejandro Pajuelo, Ramon Canal

**Affiliations:** Department of Computer Architecture, Universitat Politècnica de Catalunya, 08034 Barcelona, Spain; pol.valls@upc.edu (P.V.); beatriz.otero@upc.edu (B.O.); juan.jose.costa@upc.edu (J.J.C.); javier.verdu@upc.edu (J.V.); alex.pajuelo@upc.edu (M.A.P.); ramon.canal@upc.edu (R.C.)

**Keywords:** cybersecurity, convolutional neural network, intrusion detection systems, IoT networks, transfer learning

## Abstract

Cyberattacks in the Internet of Things (IoT) are growing exponentially, especially zero-day attacks mostly driven by security weaknesses on IoT networks. Traditional intrusion detection systems (IDSs) adopted machine learning (ML), especially deep Learning (DL), to improve the detection of cyberattacks. DL-based IDSs require balanced datasets with large amounts of labeled data; however, there is a lack of such large collections in IoT networks. This paper proposes an efficient intrusion detection framework based on transfer learning (TL), knowledge transfer, and model refinement, for the effective detection of zero-day attacks. The framework is tailored to 5G IoT scenarios with unbalanced and scarce labeled datasets. The TL model is based on convolutional neural networks (CNNs). The framework was evaluated to detect a wide range of zero-day attacks. To this end, three specialized datasets were created. Experimental results show that the proposed TL-based framework achieves high accuracy and low false prediction rate (*FPR*). The proposed solution has better detection rates for the different families of known and zero-day attacks than any previous DL-based IDS. These results demonstrate that TL is effective in the detection of cyberattacks in IoT environments.

## 1. Introduction

The rapid proliferation of IoT networks in a wide range of domains, such as manufacturing, transportation, energy, healthcare, and agriculture, among others, has interconnected billions of devices. By the end of 2022, it is expected that the number of connected devices will grow up to 46 billion. The pervasive use of IoT devices makes IoT networks vulnerable to a wide range of cyberattacks. In 2020, the leading attacks in IoT networks were worms, bots, and distributed denial of service (DDoS) [1], while in 2021 the number of cyberattacks in IoT networks doubled, according to the antivirus and security service provider Kaspersky. IoT systems are suffering devastating losses as traditional security mechanisms (e.g., traditional IDS) are too resource-demanding for IoT environments. Even worse, if we consider that IoT devices are often manufactured without appropriate security controls, a considerable percentage of IoT devices present security vulnerabilities [2].

Recent works on IoT security focus their efforts on the adoption of machine learning (ML) and deep learning (DL) techniques for intrusion detection systems (IDS). Initially, they make extensive use of ML techniques [3], but they lack the feature engineering and they have low detection rates [4]. In addition, ML-based solutions fail in the identification of different types of threats and intrusions, especially for unforeseen and unpredictable attacks. DL techniques have been subsequently adopted to overcome these constraints. They improve the ability of ML-based solutions to prevent attacks by identifying patterns that are different from normal behavior, increasing detection accuracy and reducing the false positives [5,6].

DL-based IDSs have demonstrated their capabilities to extract complex patterns when a large collection of labeled data is available to train the classification models in order to detect intrusions. However, in IoT environments, there is a lack of such large collection of labeled data for unknown (zero-day) attacks, or even for known families of attacks. In these networks, new training data are expensive and time-consuming to collect, or occasionally nonexistent. Moreover, when a new intrusion is detected, DL models must be retrained with the new data from scratch, involving a huge amount of computing resources and time. Thus, DL-based IDSs are suffering the challenges of IoT networks where datasets are scarce and unbalanced, and devices have limited computing capabilities.

The emergence of transfer learning (TL) [7] helps IDSs overcome their limitations in the detection of zero-day attacks, evolving threats, and in the effective detection of cyberattacks in networks with scarce and unbalanced datasets. TL is a recent ML progress, which applies in a target domain the knowledge previously learned in a related source domain. TL creates a high-performance learner for the target domain trained from the related source domain. TL has been demonstrated to be effective in the areas of natural language processing (NLP) [8] and computer vision (CV) [9]. Image classification models trained to detect different categories of objects are repurposed for a new, different, but related, domain. Transferring the knowledge gives better results than training the new image dataset from scratch. Research works demonstrate that the performance of a model built using TL is similar to that obtained by DL models even if the TL works with only one to ten percent of the labeled training data. Recently, TL has been explored in IDSs. It improves the detection of known attacks in domains with scarce data, such as IoT networks, and in the detection of zero-day attacks. The results are promising in the detection accuracy of new intrusions. Existing research works use TL to improve the detection of known attacks in scarce datasets; to speed up the training process; and to detect zero-day attacks. In some cases, they are focused on the detection of a specific new family of novel attacks or on a specific IoT application, such as Internet of Vehicles (IoV). Thus, this paper overcomes existing work defining a novel effective framework for the detection of different families of known and novel attacks based on TL in IoT networks.

The goal of this work is to define and implement an efficient intrusion detection framework based on TL, knowledge transfer, and model refinement. We evaluate detection rate and accuracy for known and novel cyberattack families in IoT networks with scarce and unbalanced datasets. The deep transfer learning solution developed is based on convolutional neural networks (CNNs). Two different, but related, datasets in intrusion detection are considered, containing normal and cyberattack traffic flows in the IoT domain. The BoT-IoT dataset [10] is chosen for the source domain, since it is a large dataset with IoT network traffic, and the UNSW-NB15 dataset [11] is chosen for the target domain, since it is a scarce labeled dataset with IoT network traffic that comprises modern and contemporary cyberattacks. The main contributions of this work are summarized as follows:First, we propose a novel framework for the detection of known and zero-day attacks in IoT networks, based on TL and network fine-tuning.Second, we propose the creation of three specialized datasets to train and evaluate the framework: (i) the UNSW-NB15-Basic, with normal traffic and four different types of known attacks; (ii) the UNSW-NB15-Test+, with normal traffic and five different types of zero-day attacks; and (iii) the UNSW-NB15-Test, with normal traffic and nine different types of attacks (four known and five zero-day attacks).

The proposed TL-based attack detection framework outperforms the state of the art by achieving an overall accuracy of 97.89% and 0.05% *FPR*, while detection rates for the different families of zero-day attacks range from 98.85% to 100%. The proposed framework considers IoT network traffic, such as provided in the UNSW-NB15 dataset; it does not consider real data from IoT networks.

The rest of this article is organized as follows. Section 2 discusses the related work on transfer-learning-based solutions for cyberattack detection in IoT networks. Section 3 provides the background used in our work. Section 4 presents the proposed TL-based intrusion detection framework. Section 5 and Section 6 describe the usage of the framework in IoT networks and discuss the performance evaluation results. Section 7 concludes the paper and outlines the future work.

## 2. Related Work

DL has been widely applied to network intrusion detection. However, currently available datasets in IoT environments are, in most cases, inadequate to train systems capable of detecting unknown intrusions. Transfer learning has been proposed to overcome low-level intrusion detection rates. Initial works in this area propose the use of TL [7] by means of CNN models in a two-stage learning process: first, learning from a base dataset, the UNSW-NB15 [11], and then transferring the knowledge of the learning process to the target dataset, the NSL-KDD dataset [12]. The system considers two concatenated CNNs, and it is evaluated using the KDDTest-21 dataset for considering zero-day attacks. They achieve an improvement of about 2.86% in the detection of novel attacks compared to the traditional CNN mode. They achieve an accuracy (*ACC*) of 81.94%. In the same vein, Masum et al. [13] explore transfer learning for the detection of novel intrusions. Their solution is also based on a two-step process, but in this case, the first step uses the VGG-16 pretrained on ImageNet dataset, and in the second, a deep neural network (DNN) is applied to the extracted features. The DNN consists of an input layer, two hidden layers, and an output layer. The hidden layers are fully connected layers with 64 and 8 nodes, respectively. They evaluate the solution also using the NSL-KDD dataset, achieving an accuracy of 70.97% in the detection of novel intrusions (KDDTest-21), slightly lower than [7].

Sameera et al. [14] use transfer learning in IDS to detect zero-day attacks minimizing *FPR*, but restrict the solution to the detection of remote-to-local (R2L) attacks. The system that they propose detects unlabeled R2L attacks of the NSL-KDD dataset making use of labeled DoS attacks. They achieve an accuracy of 89.79% and *FPR* of 0.15%, improving previous feature-based TL methods [15] by 11.79%. Similarly, Singla et al. [16] propose a system for detecting specific families of novel attacks transferring knowledge from a source domain to a target model with limited training data. They implement the TL model using two DNN with two and five regular densely connected layers. They break down the UNSW-NB15 dataset [11] into two parts: (i) a source dataset containing different categories of attacks, and (ii) a target dataset containing just a new attack type. The accuracy of the TL solution improves between 3.2% and 19.1%, depending on the type of new attack. The baseline is the DL model trained from scratch. In the IoV, Li et al. [17] propose the usage of TL for updating training models when a new attack is produced and the IoV cloud cannot provide the labeled data in time. Then, multiple TL is performed using the pseudo-labeled data. The experiments use two datasets of the public dataset AWID [18] and they achieve 96% detection accuracy, improving traditional schemes up to 8%. In the same area, but with a different objective, to speed up the training process, Mehedi et al. [19] propose a deep-TL-based ID model to classify normal traffic and attacks. The TL model makes use of two CNNs and the datasets used for the source and target domain are two different subsets of the new-generation labeled dataset for in-vehicle network [20], which considers three different types of attacks: flooding, fuzzing, and spoofing. The detection model shows optimal performance with an overall accuracy of 98.1%.

Fan et al. [21] bring together transfer and federated learning in 5G IoT environments. They propose a federated framework to securely enable data aggregation from different IoT networks. They use transfer learning to achieve a personalized intrusion detection model for each IoT network. They implement transfer learning using CNN, and they evaluate the solution with the CICIDS2017 [22] as base dataset and different custom target datasets for the different IoT networks. They achieve a detection accuracy of 91.93% on average.

Idrissi et al. [23] also propose the usage of transfer learning to overcome the limitation of traditional DL-based IDS on the detection of novel attacks in IoT environments with few labeled data. Their solution retrains a fine-tuned pretrained model where most of the layers are fixed and just the last ones are trained using a CNN. They consider the BoT-IoT dataset [10], generic for IoT systems, in the source domain, and update it in the target domain with small data from the TON-IoT dataset [24], specific for Industrial IoT (IIoT). They achieve an accuracy of 99% in the detection of novel attacks.

Finally, deep transfer learning (DTL) is used for intrusion detection in scarce datasets, but not considering zero-day attacks. Guan et al. [25] benefit from the work carried out in traffic classification [26] to develop a method based on deep transfer learning for network classification in IoT environments with scarce labeled data with devices with limited computing capability. They make use of EfficientNet [27] and Big Transfer (BiT) [28], which have demonstrated excellent performance for transfer learning in image recognition. They evaluate the solution using the 10% USTC-TFC2016 labeled dataset [29]. The proposal achieves an accuracy of 96.22% and 96.40% for BiT and EfficientNet, respectively. Table 1 summarizes the works reviewed in this section. Mehedi et al. [30] propose a residual neural network based on DTL to effectively detect intrusions in heterogeneous IoT networks. They construct its own dataset from various heterogeneous sources, which include seven IoT sensors, and they detect nine different types of attacks: DoS, DDoS, data injection, man-in-the-middle (MITM), backdoor, password cracking attack (PCA), scanning, cross-site scripting (XSS), and ransomware. The overall accuracy of the CNN-based model is 87%.

The work proposed in this paper differentiates from those reviewed, as some of them propose the usage of TL to improve the detection of known attacks in scarce datasets [25], heterogeneous IoT networks [30], or personalizing the solution for different IoT networks [21], but without considering novel attacks. While others are restricted to specific families of novel attacks [14,16,23] use datasets not specific to IoT environments [7,13], or focus their solution on a specific IoT environment [19]. Finally, [17] has a different objective, to speed up the training process.

## 3. Background

### 3.1. Convolutional Neural Networks

CNNs [31], also known as ConvNets, are one of the most popular DNNs. CNNs were first used for the recognition of phonemes and words [32]. Later, CNNs were applied to image classification [33]. CNNs have been recently considered in the cybersecurity field for intrusion detection [34] and encrypted traffic classification [35]. The architecture of a CNN (see Figure 1) consists of three different types of layers: convolutional, pooling, and classification. Convolutional layers are the core of the CNN, where its units are organized in feature maps. Each unit in a feature map is connected to the local patches in the feature maps of the previous layer through a set of weights (filter bank). The result of applying these filters goes through a nonlinearity transformation. The role of the pooling layers is to merge semantically similar features into a single one by applying a specific function. Pooling layers reduce the size of the feature maps and the number of overfitting parameters. The classifiers are usually formed by fully connected layers. The classification is performed based on the detected features. Recently, different pretrained models used in TL are based on CNNs [36], since the lower layers of the convolutional base of the CNN are used for general features, while the higher layers are used for specialized features.

### 3.2. Transfer Learning

DL has been successfully used in many applications with abundant training data, with the same distribution and input feature space as the testing data. However, in certain scenarios, data are scarce or expensive to collect. In such situations, it is necessary to create a high-performance learner for a target domain, with limited or no labeled examples, from a related source domain with a large collection of labeled examples. This is the fundamental purpose of transfer learning [37]. TL improves the resolution of new problems, applying the knowledge previously learned. In TL, the domains, tasks, and distributions used in training and testing can be different. For example, a model that has learned to identify cars can be used when creating a model to identify trucks. In our domain, TL is used to detect new families of attacks, applying the knowledge previously learned on the detection of existing attacks in IoT networks.

TL is formally defined in [38]. Given a source domain $D_{S}$, a source learning task $T_{S}$, a target domain $D_{T}$, and a target learning task $T_{T}$, TL contributes to improve the learning of the target predictive function $f_{T}$(.) in $D_{T}$ using the knowledge in $D_{S}$ and $T_{S}$, where $D_{S}\neq D_{T}$, or $T_{S}\neq T_{T}$.

There are two main dichotomies in the categorization of TL [38]. The first is based on the availability of labeled data, which divide TL into three categories: transductive, inductive, and unsupervised transfer learning. The second is based on the differences in feature spaces, which categorize TL into four groups: instance-based, feature-based, parameter-based, and relational-based approaches.

In the DL context, TL takes advantage of pretrained models, which, in most cases, are based on CNNs [36,39]. A typical CNN, as introduced in the previous section, has two main parts: the convolutional base for feature extraction, and the classifier. In DL models, the features computed by the first layers are general, while features extracted in the last layers are specific, more biased towards the learning task. General features can be then used in different problem domains. Yosinski et al. [40] develop the first deep transfer neural network. On this basis, three different TL strategies were conducted depending on the amount of data of the target domain. They all first substitute the original classifier by a new classifier for the target problem. Then, the resulting model is fine-tuned. However, they differentiate in the number of layers that are trained or that are fixed, ranging from training the entire model, where a large dataset is needed, to training some layers and leaving the others fixed, for small datasets with a large number of parameters. Another option is to fix the convolutional base and use its outputs to feed the classifier. It is used for small datasets or when solving really similar problems. The later approach is the chosen one in our framework since we have a small dataset without observations for the zero-day attacks and the source and target domain deal with the same problem, the detection of intrusions in IoT networks.

## 4. TL-Based Intrusion Detection Framework

This paper proposes a deep-TL-based intrusion detection framework for known and novel attacks families in IoT networks. The framework has two phases: (i) an initial *training phase on the source domain*, and (ii) a *transfer learning phase to the target domain*.

Both phases use a CNN as their learning core, CNN-B and CNN-TL, respectively, and our strategy is to fix the convolutional base and use its output to feed the classifier. Figure 2 shows the different stages of the detection process. These can be grouped into data treatment and transfer learning:Stage 1: Source domain dataset preprocessing.Stage 2: Source domain learning (CNN-B training—source dataset).Stage 3: Target domain dataset preprocessing.Stage 4: Transfer learning to the target domain (CNN-TL training—target dataset).Stage 5: Attack detection (target dataset).

### 4.1. Data Treatment and Preprocessing

Stages 1 and 3 of the TL-based intrusion detection framework accomplish the data treatment and preprocessing of the datasets, source, and target, respectively. In IoT environments, the records of network activity include traffic flows of normal and different families of cyberattacks. Raw network packets are processed to extract features and to generate the datasets. Features in IoT datasets include flow, connection, content, and time features. The types of these features usually are nominal, integer, float, timestamp, and binary. On the other hand, when considering CNN models, raw data must be converted to image format. Given all that, data treatment and preprocessing for IoT datasets usually include the following steps:One hot encoding (OHE) transformation: Transforms nominal fields to numeric using the OHE method.Decimal conversion: Converts hexadecimal fields to decimal format.Logarithmic method: Applies logarithm procedure to features with values concentrated in 0.Standardization: Standard normalization of the dataset to prevent model overfitting and biased results.Image transformation: Converts raw data to image format.

### 4.2. Transfer Learning

The TL-based framework developed for the ID framework encompasses stages 2, 4, and 5. It consists of two main phases: first, the generation of the ID-model in the source domain, and second, the update of the model, where base knowledge is transferred, to the target domain.

### 4.3. Training Phase—Source Domain

The first phase of the TL-based framework consists of the generation of the base ID-model on the source domain. To this end, it is built and trained in the source training dataset. The model accuracy is validated using the source domain validation dataset. The structure of the CNN-B is depicted in Figure 3. It consists of two convolutional layers, with 32 and 64 filters, respectively, two pooling layers to reduce the sample size of the convolution output, which contribute to the selection of most useful features, improving next step learning, and a flattening and a dense layer. The fully connected layer uses the ReLu activation function. The output layer has two outputs, with softmax activation, to determine if it belongs to normal traffic or attack.

### 4.4. Transfer Learning Phase

The second phase of the TL-based framework consists of applying the knowledge learned in the source domain to the target domain. In the CNN-TL, the convolutional base of the original base ID-model is frozen and the classifier is fed with its outputs. The convolutional base is frozen to avoid weights modification when the model is retrained. The structure of the CNN-TL model is depicted in Figure 4, and it consists of the frozen convolutional base of the CNN-B model followed by concatenated fully-connected layers as an output layer. The dropout is randomly set on a fixed probability, to reduce overfitting. The ReLu activation function is chosen in the hidden layers of the fully connected network. The number of neurons in the different layers decreases up to the output layer. The output layer contains a fully connected network with softmax activation function. The CNN-TL model is trained using the target training dataset.

## 5. Evaluation

This section proposes a possible application of the TL-based ID framework in IoT environments. We choose two datasets for the source and transfer domain that contain normal and cyberattack IoT traffic flows. For the source domain, we require a large IoT network activity dataset, then we choose the the BoT-IoT dataset [10], while for the target domain, we choose a scarce, unbalanced dataset with IoT network traffic that comprises modern and contemporary cyberattacks, the UNSW-NB15 [11] dataset. To evaluate the detection of zero-day attacks, we created four different datasets, as explained below.

### 5.1. Source Domain Dataset

In this use case, the BoT-IoT dataset [10] was chosen to train the TL framework. This dataset is created by the Australian Center for Cyber Security (ACCS), developing a realistic network environment in the Cyber Range Lab of UNSW Canberra. It consists of records of network activity in a simulated IoT environment, including both normal and several cyberattack traffic flows. It is based on the activity of a network composed of 62 hosts (based in the network mask 192.168.100.0/26). The dataset has 46 features and 5 different output classes, one for *normal traffic* and four for the different types of attacks. The dataset includes *denial of service (DoS)*, *distributed DoS (DDoS)*, *reconnaissance*, and *information theft* attacks. DoS and DDoS attacks address malicious attempts to disrupt normal traffic of a server, service, or network, overloading them with a flood of Internet traffic. DoS and DDoS attacks have the subcategories TCP, UDP, and HTTP. Reconnaissance represents attacks that gather information from the target. It has the subcategories OS fingerprinting and service scanning. Finally, information theft represents the steal of personal user information. It has the subcategories keylogging and data exfiltration. The BoT-IoT dataset comprises over 73 millions of records, but in our work, only 10% of the full dataset is considered for easy handling, maintaining the normal traffic, since it is scarce, and preserving the proportionality between the different types of attacks. Table 2 provides the details.

### 5.2. Target Domain Dataset

To validate the framework, four different datasets were derived from the UNSW-NB15 dataset [11]. This dataset is also developed by the ACCS in collaboration with researchers worldwide. It consists of normal and synthesized attack activities in a simulated IoT environment. This dataset contains normal traffic and nine different types of attacks, which include generic, exploits, fuzzers, DoS, reconnaissance, analysis, backdoor, shellcode, and worms. Table 3 provides the description and distribution (number of records) in the dataset.

The UNSW-NB15 dataset has 49 different features and it comprises over 2 million records stored in four different CSV files. The distribution between normal and malicious traffic is 87%/13%.

In order to validate the effectiveness of the TL-based ID framework in the detection of novel attacks, three different datasets are generated in the target domain:UNSW-NB15-Basic: Dataset with normal traffic and four different types of known attacks (generic, exploits, DoS, and reconnaissance) used for training. It is divided into two:–UNSW-NB15-Basic-Train: Dataset to train the initial model.–UNSW-NB15-Basic-Test: Dataset to evaluate the effectiveness in the detection of known attacks (generic, exploits, DoS, and reconnaissance).UNSW-NB15-Test+: Dataset to evaluate the effectiveness in the detection of zero-day attacks (fuzzers, analysis, backdoor, shellcode, and worms).UNSW-NB15-Test: Dataset to evaluate the effectiveness in the detection of known and zero-day attacks (generic, exploits, DoS, reconnaissance, fuzzers, analysis, backdoor, shellcode, and worms).

The first dataset created, the UNSW-NB15-Basic, is made up of normal traffic and four out of the nine types of attacks of the UNSW-NB15 dataset. The attacks considered are DoS, exploits, generic, and reconnaissance. This dataset was subsequently balanced to become closer to a real scenario, resulting in 50% of normal traffic and 50% of attacks. A total of 75% of this dataset is used for training (UNSW-NB15-Basic-Train), and the remaining 25% for testing (UNSW-NB15-Basic-Test).

The second dataset generated is the UNSW-NB15-Test+. This dataset is used to evaluate the effectiveness of the framework in the detection of unknown families of attacks. This dataset comprises normal traffic and five different types of new attacks (fuzzers, analysis, backdoors, shellcode, and worms). The new types of attacks considered are the ones in the UNSW-NB15 dataset not considered in the UNSW-NB15-Basic dataset.

The third dataset generated for evaluating the effectiveness of the framework in the detection of known and unknown families of attacks is the UNSW-NB15-Test. It consists of normal traffic and the nine types of attacks of the UNSW-NB15 dataset. Table 4 and Table 5 summarize the devised datasets.

For the source domain, we use 10% of the BoT-IoT dataset, randomly split into 75% for training and 25% for testing. Table 6 details the datasets created in terms of normal and malicious traffic. Note that the distribution of attacks is different for the UNSW-NB15-Test+ and UNSW-NB15-Test, since the former only considers zero-day attacks, while the latter considers known and zero-day attacks.

### 5.3. Data Treatment and Preprocessing

Stages 1 and 3 of the framework perform data preprocessing. First, we must bear in mind that the TL model designed uses the output of the convolutional base to feed the classifier. Then, the datasets for the source and the target domain have to be trained with the same input shape and features. To this end, we generated a new version for both datasets with their 15 common features, which are shown in Table 7.

The next step is the data preprocessing. First, the fields with a string format are transformed to numeric format using the one hot encoding (OHE) method. The fields in hexadecimal format are converted to decimal formal. The logarithm procedure is applied to fields (dur, sbytes, dbytes, and spkts) with values concentrated in 0. A standard normalization of the datasets is performed to prevent overfitting and possible biased results. Finally, data are transformed to image format, converting from 1D to 3D (i.e., vectors of length 24 results on dimension (24,1,1)).

### 5.4. Transfer Learning

First, in stage 2 of the framework, the BoT-IoT dataset is used for the generation of the base ID-model on the source domain. To this end, it is built and trained in the BoT-IoT training dataset, which represents 75% of the whole dataset. The model accuracy is validated using the BoT-IoT validation dataset, which comprises the remaining 25% of the dataset. Then, stage 4 of the framework applies the knowledge learned in the source domain to the target domain. In the CNN-TL, the convolutional base of the original base ID-model is fixed and the classifier is fed with its outputs. The CNN-TL model is trained using the UNSW-NB15-Basic-Train dataset, which represents 75% of the UNSW-NB15-Basic dataset. Finally, stage 5 of the framework performs the detection of known and zero-day attacks. To this end, the CNN-TL model is tested using the UNSW-NB15-Test and UNSW-NB15-Test+ datasets, to consider both known and novel families of attacks. Table 8 summarizes the parameters for the different classification layers of the CNN-TL model. It should be noted that in a real scenario, instead of using the UNSW-NB15-Test and UNSW-NB15-Test+ datasets, we would use the corresponding real traffic preprocessed as described in the previous section.

## 6. Results

### 6.1. Metrics

To evaluate the TL-based cyberattacks detection solution, we consider *accuracy*, *precision*, *recall*, *FPR*, and *F1-score* metrics. These metrics use properties from a confusion matrix, i.e., the matrix representation of the classification results, where true positive (*TP*) and true negative (*TN*) denote the number of attack and normal records correctly classified, whereas false positive *(FP*) and false negative (*FN*) denote the number of normal and attack records incorrectly classified.

Accuracy (*ACC*) is the ratio of correctly classified predictions over the total number of instances evaluated.
(1)$$ACC=\frac{TP+TN}{TP+TN+FP+FN}$$

Precision (*p*) is the ratio of items correctly classified from the total of items predicted.
(2)$$p=\frac{TP}{TP+FP}$$

Recall (*r*) is the ratio of items correctly classified from the total of corrected items.
(3)$$r=\frac{TP}{TP+FN}$$

False prediction rate (*FPR*) represents the ratio of items incorrectly classified (attack or normal).
(4)$$FPR=\frac{FP}{TN+FP}$$

*F1-score* is the weighted harmonic mean of precision and recall.
(5)$$F1-\mathrm{score}=\frac{2*(p*r)}{p+r}$$

#### 6.1.1. System Setup

The experimental environment is built on the Lenovo IdeaPad 320-15IKB with Intel® Core™ i7-8850U CPU @ 1.8 GHz processor and 8 GB RAM. The TL-based solution is implemented using the TensorFlow backend [41], the frontend Keras [42], Pandas, and Scikit-learn packages. The project code is uploaded and available in the GitHub repository [43].

#### 6.1.2. Training

The proposed framework is pretrained using the BoT-IoT dataset (source domain). The CNN-B is trained with 25 epochs, a batch-size of 2048, and the Adam optimizer has a learning rate of $5\times 10^{-4}$ to minimize the error function. We also use a categorical cross-entropy loss function. The CNN-TL is trained using the UNSW-NB15-Basic-Train dataset (target domain). The training consists of 15 epochs, a batch size of 4096, an Adam optimizer with a learning rate of $2\times 10^{-5}$, and categorical cross-entropy loss function. The learning rate used is very small to achieve a better performance of TL. The training parameters for both models are summarized in Table 9.

#### 6.1.3. Validation

The TL solution is tested using the UNSW-NB15-Test+ dataset (which only comprises zero-day attacks) and the UNSW-NB15-Test (which comprises both known and zero-day attacks).

First, to validate the efficiency of the TL model in the detection of zero-day attacks, we evaluate the model using the UNSW-NB15-Test+ dataset. The model achieves a 99.04% accuracy, 99.06% precision, 99.04% recall, 0.05% *FPR*, and 99.05% F1-score. Table 10 presents the number of detected samples—TP+TN, non-detected samples—FP+FN, and the detection rate for the five different types of zero-day attacks, showing that the DR exceeds 98% in all cases.

The TL model is also evaluated on the detection of known and zero-day attacks. To this end, the UNSW-NB15-Test dataset is used for testing. The CNN-TL model achieves a 97.89% accuracy, 98.22% precision, 97.89% recall, 0.05% *FPR*, and 97.97% F1-score. Although the model statistics are slightly lower than when only considering zero-day attacks, the detection rate for the different families of zero-day attacks is also greater than 98%, as shown in Table 11.

To draw a comparison with existing DL-based IDS, we implement a solution based on CNN and compare its performance with our proposed TL-based solution. The CNN model is evaluated first on the detection of zero-day attacks with the NSW-NB15-Test+ dataset. It achieves an overall accuracy of 71.85%, a precision of 80.77%, a recall of 71.85%, an *FPR* of 63.64%, and an F1-score of 68.14%. We also evaluate the CNN model for the detection of both known and zero-day attacks using the UNSW-NB15-Test+ dataset. In this case, the accuracy is 85.38%, precision is 91.44%, recall is 85.38%, *FPR* is 16.04%, and the F1-score is 87.29%. If we compare the overall metrics for both solutions, we can conclude that the TL model outperforms the CNN model for all metrics.

Accuracy is improved by 27.19% in the detection of zero-day attacks, and 12.51% when considering both known and zero-day attacks. The main reason for this improvement is that the optimizing algorithm converges faster to optimal weights when the CNN-TL model has weights initialized on a similar domain, unlike the CNN model that starts from scratch, in our case using random weights.

Another metric that improves significantly is *FPR*. The non-transfer-learning-based solution has higher *FPR* values due to two main factors: the occurrence of zero-day attacks and the imbalanced nature of the dataset. From these experiments, we can conclude that TL helps reduce false positives in intrusion detection, an important concern in the industry of cyberattack detection systems.

Finally, we analyze the detection rate (DR) for each family of attacks. We consider two IDSs—the CNN-based and the TL-based—and for the two scenarios: (i) considering *only zero-day attacks* (UNSW-NB15-Test+), and (ii) considering *both known and zero-day attacks* (UNSW-NB15-Test).

Table 12 shows the complete picture from which we can draw the following conclusions. The TL-based IDS has better detection rate for all the different types of attacks, known and zero-day. The DR achieved for the different families of zero-day attacks remains practically the same when only considering zero-day attacks, or when also considering known attacks, ranging from 98.85% to 100%. When compared with the CNN-based IDS, the TL-based framework improves DR for zero-day attacks up to 33.28%, and up to 7.1% for known attacks. Therefore, the TL-based IDS significantly improves the detection of zero-day and known attacks with less representation in the dataset, when compared with the CNN-based IDS.

## 7. Conclusions and Future Work

In this paper, we investigate the feasibility of deploying transfer-learning-based intrusion detection for zero-day attacks in IoT networks with scarce and unbalanced datasets. To this end, we develop an efficient intrusion detection framework that combines knowledge transfer and model refinement, with excellent detection accuracy for known and novel cyberattacks families. We implement the solution considering the BoT-IoT dataset to learn the knowledge (source domain) and applying it onto the UNSW-NB15 dataset (target domain). In order to evaluate the TL-based ID framework for zero-day attacks, we generate a test dataset with five different types of novel attacks. We find that transfer learning and network fine-tuning improve IDS even in unbalanced datasets with enough labeled data for the detection of zero-day attacks. The experimental results show that the TL-based framework achieves an excellent accuracy and a very low *FPR*. DRs significantly improve for the different families of known and novel attacks, compared to previous DL-based IDS. The proposed framework considers IoT network traffic, provided in the UNSW-NB15 dataset, and future work will consider real data from IoT networks, as well as extending the proposed solution on the detection of other types of zero-day attacks, but also evaluating its performance on lightweight IoT devices with real IoT network traffic data.

## Figures and Tables

**Figure 1 sensors-22-05621-f001:**
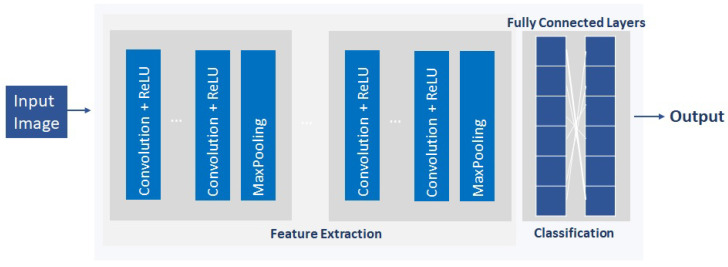
CNN architecture.

**Figure 2 sensors-22-05621-f002:**
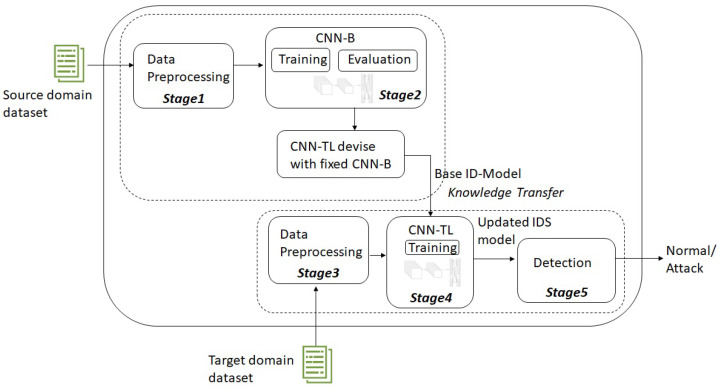
Overall structure of the proposed intrusion detection framework.

**Figure 3 sensors-22-05621-f003:**
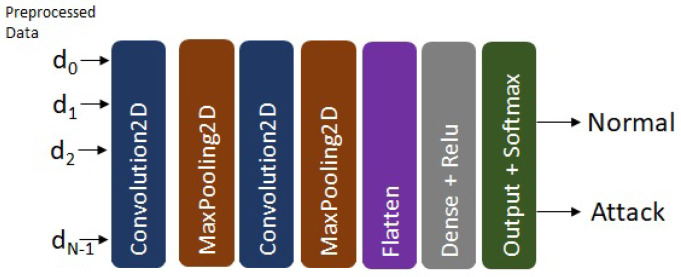
CNN-based IDS-model structure.

**Figure 4 sensors-22-05621-f004:**
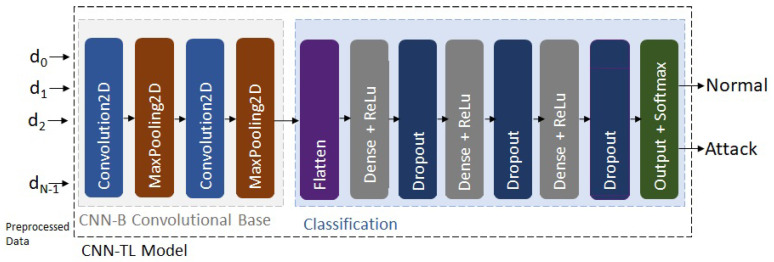
CNN-TL IDS-model structure.

**Table 1 sensors-22-05621-t001:** Transfer-learning-based intrusion detection in IoT environments.

Reference	TL	Source Dataset	Target Dataset	Accuracy
Wu et al. [7] (2019)	CNN-CNN	UNSW-NB15	NSL-KDD	81.94%
Masum et al. [13] (2020)	DNN-DNN	VGG-16	NSL-KDD	70.97%
Sameera et al. [14] (2019)	PCA-KNN	NSL-KDD (DoS+ Normal)	NSL-KDD (R2L+ Normal)	89.79%
Singla et al. [16] (2019)	DNN-DNN	UNSW-NB15 Subset	UNSW-NB15 Subset (single new attack)	95–98%
Li et al. [17] (2021)	SVM-RF	AWID	AWID	96%
Mehedi et al. [19] (2021)	CNN-CNN	Custom	Custom	98.1%
Fan et al. [21] (2021)	CNN-CNN	CICIDS2017	Custom	91.93%
Idrissi et al. [23] (2021)	CNN-CNN	BoT-IoT	TON-IoT	99.43%
Guan et al. [25] (2021)	BiT EfficientNet	Custom	10% USTC-TFC2016	96%
Mehedi et al. [30] (2022)	CNN	Custom	Custom	87%

**Table 2 sensors-22-05621-t002:** BoT-IoT Dataset.

Category	Subcategory	Records	Description
Normal	Normal	9543	Natural transaction data.
DoS	TCP UDP HTTP	38,532,480	A malicious attack to cripple the services offered by a site, server, or network overloading the target of its associated infrastructure by flooding the site with many requests.
DDoS	TCP UDP HTTP	33,005,194	Attack where multiple compromised computer systems attack a target, causing a DoS.
Reconnaissance	OS fingerprinting Service scanning	1,821,639	All the different strikes simulating attacks gathering information.
Information Theft	Keylogging Data exfiltration	1587	Stealing of personal user information.

**Table 3 sensors-22-05621-t003:** UNSW-NB15 dataset.

Category	Records	Description
Normal	2,218,761	Natural transaction data.
Generic	215,481	Attack against blockciphers with a given block and key size (not considering its structure).
Exploits	44,525	Attack that exploits vulnerabilities, taking advantage of security problems (of an operating system or a piece of software) known by the attackers.
Fuzzers	24,246	Attack that suspends a program or network, feeding it with randomly generated data.
DoS	16,353	A malicious attack that makes a server or network resource unavailable, overloading the target of the associated infrastructure with a flood of Internet traffic.
Reconnaissance	13,987	Comprises different attacks that gather information.
Analysis	2677	Different attacks on penetrations (HTML files, spam, and port scan).
Backdoors	2329	An attack that bypasses a system security mechanism to access a computer or its data.
Shellcode	1511	Attack that exploits software vulnerabilities using small pieces of code as payloads.
Worms	174	Attack where the attacker replicates itself to spread to other computers.

**Table 4 sensors-22-05621-t004:** UNSW-NB15-Basic-Train and UNSW-NB15-Basic-Test datasets.

	UNSW-NB15-Basic-Train		UNSW-NB15-Basic-Test	
Name	Records	Percentage	Records	Percentage
Normal	217,552	49.95%	72,794	50.14%
Generic	161,865	37.17%	53,616	36.93%
Exploits	33,408	7.67%	11,117	7.66%
DoS	12,196	2.80%	4157	2.86%
Reconnaissance	10,498	2.41%	3489	2.40%

**Table 5 sensors-22-05621-t005:** UNSW-NB15-Test+ and UNSW-NB15-Test datasets.

	UNSW-NB15-Test+		UNSW-NB15-Test	
Name	Records	Percentage	Records	Percentage
Normal	30,937	50.00%	321,283	50.00%
Generic	-	-	215,481	33.53%
Exploits	-	-	44,525	6.93%
DoS	-	-	16,353	2.54%
Reconnaissance	-	-	13,987	2.18%
Fuzzers	24,246	39.19%	24,246	3.77%
Analysis	2677	4.33%	2677	0.42%
Backdoor	2329	3.76%	2329	0.36%
Shellcode	1511	2.44%	1511	0.24%
Worms	174	0.28%	174	0.03%

**Table 6 sensors-22-05621-t006:** Dataset summary showing the number of records corresponding to normal and malicious traffic, the corresponding percentage of attacks, and the percentage of novel attacks.

Dataset	Normal	Attack	% Attack	% Novel Attack
BoT-IoT	9543	5,823,226	99.84%	-
UNSW-NB15-Basic-Train	217,552	217,967	50.04%	-
UNSW-NB15-Basic-Test	72,794	72,379	49.85%	0.00%
UNSW-NB15-Test+	30,937	30,937	50.00%	100.00%
UNSW-NB15-Test	321,283	321,283	50.00%	9.63%

**Table 7 sensors-22-05621-t007:** Common features for the BoT-IoT and UNSW-NB15 datasets.

	BoT-IoT	UNSW-NB15	Type	Description
1	proto	proto	nominal	Textual representation of transaction protocols present in network flow.
2	saddr	srcip	nominal	Source IP address.
3	sport	sport	integer	Source port number.
4	daddr	dstip	nominal	Destination IP address.
5	dport	dsport	integer	Destination port number.
6	spkts	spkts	float	Source-to-destination packet count.
7	dpkts	dpkts	float	Destination-to-source packet count.
8	sbytes	sbytes	float	Source-to-destination byte count.
9	dbytes	dbytes	float	Destination-to-source byte count.
10	state	state	nominal	Transaction state.
11	stime	stime	timestamp	Record start time.
12	ltime	ltime	timestamp	Record last time.
13	dur	dur	float	Record total duration.
14	attack	label	binary	Class label: 0 for normal traffic, 1 for attack.
15	category	attack_cat	nominal	Cyberattack family.

**Table 8 sensors-22-05621-t008:** Classification layers parameters summary: CNN-TL model.

Classification Head	Layer 1	Layer 2	Layer 3	Output Layer
Number of neurons	448	224	112	2
Dropout probability	0.4	0.3	0.3	-
Activation	ReLu	ReLu	ReLu	Softmax

**Table 9 sensors-22-05621-t009:** TL model training parameters summary.

Model	Epochs	Batch Size	Optimizer	Learning Rate	Loss
CNN-B	25	208	Adam	$5\times 10^{-4}$	Categorical cross-entropy
CNN-TL	15	4096	Adam	$2\times 10^{-5}$	Categorical cross-entropy

**Table 10 sensors-22-05621-t010:** Attack detection summary UNSW-NB15 dataset: Zero-day attacks.

Traffic	Detection Rate	Detected Samples	Non Detected Samples
Normal	98.34%	30,358	513
Analysis	100.00%	622	0
Backdoor	100.00%	357	0
Fuzzers	99.95%	21,507	10
Shellcode	99.93%	1510	1
Worms	98.85%	172	2

**Table 11 sensors-22-05621-t011:** Attack detection summary UNSW-NB15 dataset: Known and zero-day attacks.

Traffic	Detection Rate	Detected Samples	Non Detected Samples
Normal	98.53%	315,902	46,081
DoS	99.43%	3841	22
Exploits	99.75%	28,249	68
Generic	99.98%	213,678	40
Reconnaissance	99.94%	11,848	6
Analysis	99.84%	621	1
Backdoor	99.44%	355	2
Fuzzers	99.79%	21,472	45
Shellcode	99.93%	1510	1
Worms	98.85%	172	2

**Table 12 sensors-22-05621-t012:** Attack detection rate for known and zero-day attacks.

	UNSW-NB15-Test			UNSW-NB15-Test+		
Traffic	CNN	TL	Improvement	CNN	TL	Improvement
Normal	99.65%	98.54%	−1.11%	98.52%	98.34%	−0.18%
DoS	96.73%	99.43%	2.7 0%	-	-	-
Exploits	97.90%	99.76%	1.86%	-	-	-
Generic	99.16%	99.98%	0.82%	-	-	-
Reconnaissance	92.85%	99.95%	7.10%	-	-	-
Analysis	86.14%	99.84%	13.7%	66.72%	100.00%	33.28%
Backdoor	83.62%	99.44%	15.82%	89.64%	100.00%	16.38%
Fuzzers	80.76%	99.79%	19.03%	69.20%	99.95%	30.75%
Shellcode	89.43%	99.93%	10.50%	98.34%	99.93%	1.59%
Worms	96.31%	98.85%	2.54%	95.97%	98.85%	2.88%

## Data Availability

The TL-based IDS framework is available at the GitHub repository github.com/polvalls9/Transfer-Learning-Based-Intrusion-Detection-in-5G-and-IoT-Networks.

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
