# Peer review of "Transfer-Learning-Based Intrusion Detection Framework in IoT Networks"

_sensors, 2022, doi:10.3390/s22155621_

Round 1
Reviewer 1 Report
This is a nicely written and organized paper. It has motivated and introduced the problem well. It has also given a thorough overview of the background machine learning concepts that the proposal relies on.
Unfortunately, as neat as the paper may be from the machine learning point of view, it is doubtful that it contributes to the solution of the attack detection problem in IoT since there is a plethora of heterogeneous, superposed, moving IoT instances in real scenarios.
In such a scenario, the proposed approach would not help tackle the problem because the proposed TL model does not present the means for collecting and analyzing the multiple IoT data flows. Moreover, the paper does not show any performance figure to ensure that the IDS will intruders before they dominate the IoT devices and software artifacts.
In that sense, it seems that this paper proposal is more a computational tool for analyzing a previously recorded database of attacks than an intrusion detection system for an existing IoT scenario.
I suggest the paper clarify the IDS architecture showing the IDS deployment in heterogeneous IoT network instances and the data collection from the distributed IoT modules. It would be interesting to evaluate how the proposed IDS interacts with the IoT devices, the network infrastructure, and the IoT software components. Furthermore, studying the interactions with intermediation modules, gateways, distributed middleware, and cloud applications is interesting to evaluate how the TL modules operate in a more realistic scenario.
Reviewer 2 Report
Since the source domain and target domain datasets have different attack distributions, the reviewer doesn't believe that the transfer-learning-based IDS design can work. Specifically, the BOT-IOT dataset includes most data in attack type. By contrast, UNSW-NB15 includes most data is normal. The diversity of attack types in UNSW-NB15 is much more than BoT-IoT (DDoS only). Therefore, to build a profile (assume that it is the attack) from BoT to transfer for usage in UNSW-NB15 is theoretically wrong. If building an attack profile, the source dataset must have wide coverage. Also, if making a normal profile, the source dataset must have a massive number of normal data points.
Since the fundamental concepts are misleading, the whole building and testing results cannot be reliable. The authors are encouraged to clarify the architecture design and justify the model parts by visualization.
Reviewer 3 Report
This paper proposes a Transfer Learning (TL) based IDS for the effective detection of zero-day attacks, tailored to 5G IoT scenarios with unbalanced and scarce labeled datasets. The manuscript has merit but requires a major review for being accepted. However, there are some suggestions to be revised to improve the quality of this paper as follows:
1. Abstract should be clear and check whether all the points mentioned in abstract are addressed in this manuscript. The abstract must summarize the performance evaluation results.
2. The authors should provide solid motivation for their work based on the existing literature. In addition, the main contributions should be defined as set of bullets at the end of introduction section.
3. A table should be added to summarized related work and state the approach and result achieved by other research in the introduction.
4. The paper contains only references before 2022; other years references also need to be added especially of 2021 and 2022.
5. English usage needs to be improved. Please carefully proofread the paper to get rid of typos and grammar mistakes.
6. References should follow the template. Please correct them all.
Reviewer 4 Report
In this paper, the authors propose a Transfer Learning (TL) based IDS for the effective detection of zero-day attacks, tailored to 5G IoT scenarios with unbalanced and scarce labeled datasets. Moreover, evaluation work was performed.
To improve the quality of the paper, a few suggestions are as follows.
1. The literature review in Part background and related work is insufficient and a little outdated. Please improve it using recent papers.
[1] S. T. Mehedi, A. Anwar, Z. Rahman, K. Ahmed, and I. Rafiqul, "Dependable Intrusion Detection System for IoT: A Deep Transfer Learning-based Approach," IEEE Transactions on Industrial Informatics, 2022.
[2] Y. Fan, Y. Li, M. Zhan, H. Cui, and Y. Zhang, "Iotdefender: A federated transfer learning intrusion detection framework for 5g iot," in 2020 IEEE 14th International Conference on Big Data Science and Engineering (BigDataSE), 2020, pp. 88-95: IEEE.
[3] X. Li, Z. Hu, M. Xu, Y. Wang, and J. Ma, "Transfer learning based intrusion detection scheme for Internet of vehicles," Information Sciences, vol. 547, pp. 119-135, 2021.
[4] S. T. Mehedi, A. Anwar, Z. Rahman, and K. Ahmed, "Deep transfer learning based intrusion detection system for electric vehicular networks," Sensors, vol. 21, no. 14, p. 4736, 2021.
2. Figs 1 and 4 is vague. Pls redraw it.
3. A few variables are not defined. Please correct it.
4. Evaluation Platform and environment settings should be described in detail since the performance of the proposed method are evaluated wholly based on the specific experiments.
6. Rewrite the conclusion section in the summarized form. Rewrite the abstract part to tell people what is the main contribution of this paper.
7. Some sentences should be rewritten and some grammar mistakes should be corrected.
Round 2
Reviewer 2 Report
The reviewer found no information about the merged dataset (combining BoT-IoT and UNSW-NB15) in Table 6.
The authors may need to explain how such a method can run through the shared code in [43].
Reviewer 3 Report
In my opinion, the manuscript may be accepted. However, some minor changes are needed.
First, some variables in the mathematical framework are not introduced or described in the text. That makes difficult to follow your proposal.
Besides, several format errors are all across your paper. Please, review the manuscript carefully.
Reviewer 4 Report
this paper can be accepted for publication in this journal.
